# Natal and Neonatal Teeth: A Case Report and Mecanistical Perspective

**DOI:** 10.3390/healthcare8040539

**Published:** 2020-12-04

**Authors:** Emil Anton, Bogdan Doroftei, Delia Grab, Norina Forna, Mihoko Tomida, Ovidiu Sebastian Nicolaiciuc, Gabriela Simionescu, Eugen Ancuta, Natalia Plopa, Radu Maftei, Ciprian Ilea, Carmen Anton

**Affiliations:** 1Department of Obstetrics and Gynecology, University of Medicine and Pharmacy “Grigore T. Popa”, 700115 Iasi, Romania; emil.anton@yahoo.com (E.A.); bogdandoroftei@gmail.com (B.D.); ovidiunicolaiciuc@yahoo.com (O.S.N.); gabi.ginecologie@gmail.com (G.S.); plopa_nati@yahoo.com (N.P.); dr.radu.maftei@gmail.com (R.M.); cilea1979@yahoo.com (C.I.); carmen_ro2008@yahoo.com (C.A.); 2Clinical Department, Clinical Hospital of Obstetrics and Gynecology “Cuza Voda”, 700038 Iasi, Romania; 3Origyn Fertility Center, 700032 Iasi, Romania; 4Department of Oral Health Promotion, Graduate School of Oral Medicine, Matsumoto Dental University, Nagano 399-0781, Japan; mihoko.tomida@mdu.ac.jp; 5Clinical Department, Spital Clinic Obstetrica Ginecologie “Elena Doamna Iasi”, 700398 Iasi, Romania; eugen01ro@yahoo.com; 6Clinical Department, Sf. Spiridon Clinical Hospital, nr.1, 700111 Iasi, Romania

**Keywords:** natal teeth, neonatal teeth, case report, dentition, extraction

## Abstract

The presence of teeth on babies earlier than four months is a rare condition. Therefore, adequate treatment for each case should be instituted as soon as possible, considering that certain complications may arise. This report describes a rare case in which a newborn baby required the extraction of two mobile mandibular natal teeth to prevent the risk of aspiration. After two years, the clinical re-evaluation showed a residual tooth instead of a temporary one. This case report shows that adequate diagnosis should include a radiographic examination to determine whether these teeth are components of normal or supernumerary dentition, as well as further investigations on the relationship with the adjacent teeth. Another important aspect highlighted in this case report is the need for a post-extraction curettage of the socket in order to reduce the risk of ongoing development of the dental papilla cells.

## 1. Introduction

Natal and neonatal teeth, also known as dentitia praecox, dens connatalis, congenital teeth, fetal teeth or precocious dentition, have been the subject of research since 59 BC, when Titus Livius first reported such a case [1].

However, the terms natal or neonatal teeth were used for the first time by Massler and Savara in 1950 to define teeth that erupt before or a few months after birth [2]. For a distinction of the two terms, natal teeth are those teeth that are present at birth. On the other hand, neonatal teeth are those teeth that develop in the individual’s first month of life. In addition, the teeth that erupt within one and three and a half months of life are called early infancy teeth. Natal teeth are more common, with an approximate ratio of 3:1 more frequent occurrence compared to neonatal teeth.

Although the etiology of this condition is unknown, researchers have incriminated various hypothetical factors. One of the proposed factors is represented by pollutants such as polychlorinated biphenyls (PCBs), dibenzofurans (PCDFs) and polychlorinated dibenzo-*p*-dioxins (PCDDs) [3]. Authors have suggested that these pollutants have the capacity to cross the placenta. Furthermore, concentrations of PCDDs, PCBs and PCDFs that were found in the adipose tissue of a newborn child have also been detected in the milk of the mother [4].

The incidence of birth and neonatal teeth is extremely reduced and is documented to widely vary, from 1:2000 to 1:3500, with one or two teeth the most common occurrence [5]. The majority of researchers report the absence of gender predilection, although according to some studies, this condition is slightly more common among females [6,7].

The most common localization of natal/neonatal teeth is the mandibular region of central incisors (85%), consistent with the normal order of eruption of primary deciduous teeth, followed by maxillary incisors (11%), mandibular cuspids or molars (3%) and then maxillary cuspids or molars (1%) [8]. Natal/neonatal teeth may vary in size, shape or color, from small to normal size, from conical to normal shape and from yellowish to opaque color. Usually, these teeth have an immature appearance with enamel hypoplasia and dentin, with small or absent root formation. Most of the natal/neonatal teeth are mobile, due to a poor attachment to the alveolar ridge through a pad of soft tissue. Moreover, the histological aspect of a natal/neonatal tooth shows a thin enamel layer, with varying degrees of mineralization, or a total absence of enamel in some regions. These teeth are doubled or in pairs in 61% of cases and the development of succedaneous teeth is normal [9].

Thus, for each newborn baby a complete oral examination is required for early diagnosis and treatment, so that possible complications may be avoided. The most serious complication that may occur is the aspiration of the mobile tooth followed by asphyxia of the infant. In addition, most of these teeth erupt from the normal complement of primary teeth (90–99%) and only 1–10% of them are supernumerary.

Therefore, natal/neonatal teeth can be categorized using Helbing’s classification, based on their clinical characteristics. The first category includes natal/neonatal teeth with a shell-shaped crown that is poorly attached to the alveolus by a rim of oral mucosa and absence of a root. The second category contains natal/neonatal teeth that present a solid crown poorly attached to the alveolus by oral mucosa, with little or no root at all. The third category consists of natal or neonatal teeth that have an incisal edge of the crown just erupted through the oral mucosa. Lastly, the fourth category is formed by unerupted but palpable teeth with only visible mucosal swelling [10].

Given the relative rareness of natal/neonatal teeth occurrence and the possible serious complications that may arise, the main purpose of the present paper was to present the case of a newborn baby with natal teeth in the mandibular anterior region. By presenting our case report and analyzing the available literature, we believe we emphasize the importance of a comprehensive and detailed oral cavity exam for each newborn baby.

## 2. Case Report

A female newborn baby was thoroughly examined by a neonatologist at Iași “Cuza Vodă” Maternity Hospital, six hours after birth, due to difficulty of breastfeeding and presence of two tooth-like structures on the lower jaw. According to the data, the patient was delivered at full term, vaginally, without any perinatal complication, weighing 3700 g at birth. The pregnancy was monitored from the beginning, without any complications. Except obesity (BMI: 35.4 kg/m^2^) and polycystic ovary syndrome, the mother of the infant had no other personal pathological history. Furthermore, the mother did not take any other medication during pregnancy except for vitamins. No other relative had natal/neonatal teeth. All ethical aspects regarding this case were strictly respected: written consent was obtained from both parents regarding participation in the present case report.

The examination of the baby’s oral cavity revealed two tooth-like structures covered by gingival tissue, in the anterior mandibular region, at the lower central incisor position (Figure 1).

The remaining gum pads, tongue and intraoral mucosa appeared normal. According to Miller’s classification of mobility, the clinical condition was natal teeth with grade II mobility, while the appearance was categorized according to Hebling’s classification into category number 4. Two days later, the newborn was referred to the Department of Oral and Maxillofacial Surgery, Sf. Spiridon University Hospital and Gr. T. Popa University of Medicine and Pharmacy, Iasi, Romania. After a complete examination, the care plan proposed by the medical team included a radiographic exam of the gingival area and teeth extraction, taking into consideration Miller’s classification and the high risk of aspiration.

Subsequently, the plan of treatment was comprehensively explained to the parents, but written consent was obtained only for the extraction and not for the radiographic exam. The two natal teeth were extracted with extraction forceps under topical anesthesia which was well tolerated by the patient. The shape, size and color of the two teeth were similar to normal teeth (Figure 2).

Curettage of the socket was performed to prevent ongoing development of cells of the dental papilla (Figure 3). Postoperatively, there were no complications such as bleeding or infection. Instructions of oral hygiene for the newborn baby were given to the mother and during re-evaluation two days later, the wound was found to be healed. Therefore, the mother was allowed to resume breastfeeding.

The child developed normally, however, the two-year post-extraction examination revealed the presence of all age-specific teeth with the exception of the mandibular central incisors, suggesting that the extirpated natal teeth were actually part of temporary dentition. Additionally, instead of the lower left temporary central incisor, a residual tooth-like structure with solid and immobile aspect was observed, most likely due to the ongoing development of dental papilla cells. Furthermore, it could be seen that a tooth-like structure instead of the lower left temporary central incisor, and no appreciable space loss, occurred following the extraction (Figure 4). A summary of the case report can be seen in Table 1.

## 3. Discussion

The presence of teeth on newborn babies is uncommon; therefore this phenomenon is linked to various myths in many cultures. For example, in British culture, it is believed that infants born with natal teeth would become popular warriors, whereas those infants are considered to be bearers of misfortune in China, India and Poland. Furthermore, they are predicted to be the world’s potential conquerors in France and Italy, because, apparently, historical figures such as Napoleon Bonaparte and Julius Caesar were born with this condition [11].

This specific condition is usually diagnosed at birth during the general inspection performed by the neonatologist. In regards to the causes and risk factors associated with natal teeth, there is no evident or established single cause. Researchers have incriminated various hypothetical causes such as infection (congenital syphilis), osteoblastic activity in the germ region, congenital syndromes, maternal exposure to environmental toxins (polychlorinated biphenyls (PCB) and dibenzofurans), febrile episodes during pregnancy and nutritional deficiency (hypovitaminosis) [12]. Another cause of this condition is thought to be endocrine disturbances in the mother’s body due to excessive secretions of the pituitary, thyroid or gonad. Yet, the most acceptable theory is the one postulated by Hal in 1957, according to which presence of natal/neonatal teeth is due to superficial position of the tooth germ [13]. This implies that the tooth is not located in an alveolus, but below the surface of the alveolar bone, above the germ of the permanent successor. Furthermore, this theory was demonstrated by Boyd and Miles on the mandibular anatomical sections of a stillborn fetus [14]. Therefore, this unusual location predisposes the tooth to erupt earlier and is related to a hereditary factor [15]. Another paper that confirmed this hypothesis was a study led by Kates et al. in 1984 [16]. These researchers found that 7 out of 38 infants with natal/neonatal teeth had a positive family history of natal or neonatal teeth: two siblings, one mother, three paternal grandfathers and one paternal grandmother.

On the other hand, Štamfelj et al. examined the size, ultrastructure and microhardness of two natal teeth without permanent successor germs, extracted from a female patient when she was 7 years old [17]. This group of researchers compared the characteristics of both natal teeth and normal primary teeth, and concluded that the natal teeth were prematurely erupted regular primary mandibular incisors. They related the occurrence of natal teeth, associated with agenesis of their primary successors, to an accelerated or premature pattern of dental development, rather than to superficial positioning of the teeth germs [17].

Regarding the shape of natal/neonatal teeth, they are frequently smaller and more conical compared to normal temporary teeth [18]. Additionally, their color is often yellow or white but they might represent hypoplastic enamel accompanied by poorly developed or even absent root [5]. A conical shape has been found in 40% of the natal/neonatal teeth and hypoplastic enamel and dentin in 10% of the natal/neonatal teeth [19]. The presence of hypoplastic enamel on natal/neonatal teeth might be explained by immature enamel which is not able to complete its development as soon as the gingival coating disappears and subsequently the underdeveloped enamel starts deteriorating [16]. Specifically, when the enamel becomes unprotected, it generally turns into a yellow–brown color and keeps breaking down as long as it remains exposed [20]. Moreover, the available data from the literature show that the natal/neonatal teeth come in pairs in different percentages varying from study to study, from 38% to 43.3% to 76% [5,16,20] The shape, size and color of the two natal teeth in our case report were similar to normal teeth with no hypoplastic enamel and dentin. The plausible reason why the teeth from our case report were whitish is that these teeth were protected by the gingival coating until extraction.

Regarding the gender differences, the available literature shows that more females are affected by natal/neonatal teeth (63.3%) compared to males [16,21,22] However, no significant differences between males and females are found in the literature regarding natal/neonatal tooth morphology, positive family history or complications [19].

Moreover, in the literature, numerous congenital syndromes are considered to be associated with natal/neonatal teeth [23,24,25]. A thorough examination of the available literature shows that the most common congenital syndromes correlated with natal or neonatal teeth are: Ellis–van Creveld, Jadassohn–Lewandowsky, Hallermann–Streiff, Rubinstein-Taybi, steatocystoma multiplex, Pfeiffer, craniofacial dysostosis and adrenogenital syndromes [23,24,25]. In addition, natal and neonatal teeth are more frequent in children with cleft lip than among the general population [26]. It is important to be mentioned that the patient from our case report was not associated with any anomalies or syndromes.

The most common complication of natal teeth is the ulceration of the tongue’s ventral surface, or Riga–Fede disease, which is the result of repetitive trauma of the area [27]. This condition leads to difficulty in feeding or refusal to feed because of the pain. Other complications may include: potential risk of swallowing and aspiration of tooth, due to its great mobility, injuries to the mother’s breast and apical abscesses.

Therefore, it is essential to perform a radiographic examination to establish if these natal/neonatal teeth are components of normal dentition or are supernumerary, to determine the amount of root development and to establish differential diagnosis. Bohn’s nodules and congenital epulis might be confused with natal teeth. Another advantage of radiography is that it may reveal the relationship between the natal teeth and adjacent teeth. The care plan should be formulated with proper regard for the need to ensure normal dental occlusion.

In addition, factors such as implantation and degree of mobility, breastfeeding interference and the possibility of dislocation followed by swallowing or aspiration of these teeth should be taken into consideration when establishing the treatment plan. Over the years, various management methods of natal teeth have been proposed. For example, Padmanabhan et al. recommended in 2010 that the incisal edges of the natal or neonatal teeth to be grinded or smoothed with an abrasive instrument, to prevent the injury of the maternal breast [28]. Moreover, Choi et al. reported rapid healing of a sublingual traumatic ulceration after applying composite resin over the affected teeth [29]. However, this procedure might be limited due to a reduced surface area of enamel available for resin bonding and the expected lack of cooperation from the infant.

Consequently, extraction of these teeth is preferred and should be performed if the teeth are supernumerary or if the teeth are extremely mobile, due to the potential complications mentioned above. However, some authors suggest that natal teeth should be extracted only if they belong to Hebling’s category number 1 or 2 and the degree of mobility is more than 2 mm [30]. According to Hebling and Zuanon, if the teeth are components of normal dentition, premature extraction may cause a loss of space and collapse of the developing mandibular arch. Contrary to this, in a study conducted in Hong Kong on 48 children, 56 out of 72 natal teeth were removed with no appreciable space loss occurred following the extractions [31]. Due to high mobility, extraction of these teeth may be done with a forceps or even with the fingers. The fourth category of natal teeth in Hebling’s classification, represented by natal teeth covered by gingival mucosa, is less likely to generate complications such as aspiration or swallowing of the teeth. Although, during breastfeeding the thin layer of gingival mucosa may get traumatized, exposing the mobile teeth and therefore may increase the risk for the tooth to be dislodged and inhaled or swallowed by the infant.

Another important risk is represented by possible local hemorrhage. Furthermore, the risk of hemorrhage is directly proportional to the degree of mobility of the natal teeth, so the more mobile the tooth is, the lower the risk of hemorrhage. However, The American Academy of Pediatrics advises that a single intramuscular dosage of 0.5 to 1 mg of vitamin K should be administered to all newborn babies before extraction, since it is essential for the formation of clots at the extraction site [32]. In Romanian standard protocol, and in our presented case reports, every newborn receives a prophylactic dose of vitamin K a few hours after birth. Usually, this vitamin is synthesized by bacteria in the large intestine and since commensal intestinal flora will not have been formed until the infant is 10 days old, extraction below this age is regularly associated with an increased hemorrhage risk. However, it is the doctor’s choice whether to administer prophylactic vitamin K before extraction or not, although it is standard procedure in most of the hospitals to perform a single intramuscular dose of vitamin K immediately after birth [33].

The extraction of the natal teeth should be followed by a gentle curettage of the socket to remove the underlying dental papilla and Hertwig’s epithelial root sheath [34]. Failure to curette the socket may cause an ongoing development of the cells of the dental papilla, which may result in eruption of tooth-like structures several months later, referred to by Tsubone et al. as “residual natal tooth” [35]. According to King and Lee (1998), the risk of residual tooth formation is approximately 9.1%. In these cases if this residual tooth formation develops, a second surgical procedure is required [2]. In addition, lately there is a tendency for much clearer management in the cases of prematurely erupted teeth in newborns [36].

These aspects could be also relevant in the recent contexts regarding the connections between dental, neuropsychiatric and gastrointestinal processes, having for example oxidative stress and other complex factors in the center of these manifestations [36].

Regarding the limitations of our study, we can mention here the lack of intra-oral radiographs, since no written consent was obtained for the radiographic exam.

## 4. Conclusions

A comprehensive and detailed oral cavity exam is required to be performed on each newborn baby. Early diagnosis and appropriate care is important, as birth teeth may affect the child’s physical growth, but this may also have a psychological impact on both the child and the parents. Conservative care, followed by good dental hygiene, is recommended. Extraction is the safest option if there are complications or if the teeth are mobile. When using topical anesthesia, extraction should be followed by gentle, but firm and complete, curettage of the socket. In both scenarios, periodic follow-up by a pediatric dentist is needed to prevent the occurrence of early childhood caries, to supervise the development of the future dentition and to manage an eventually residual natal tooth. Further studies are required in order to understand the etiology and nature of natal and neonatal teeth.

## Figures and Tables

**Figure 1 healthcare-08-00539-f001:**
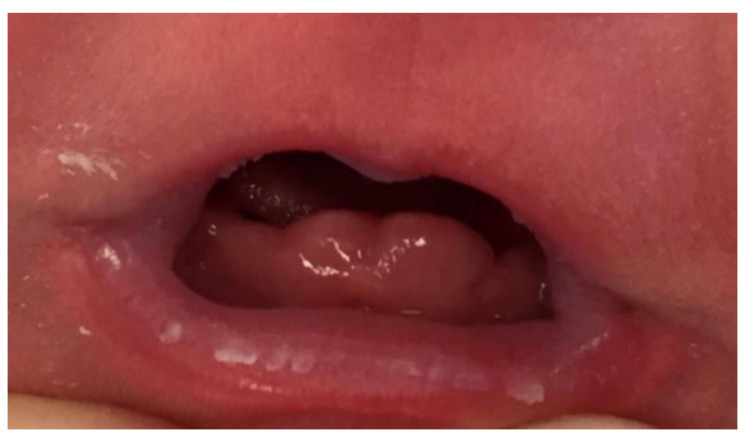
Two-day-old female infant with natal teeth.

**Figure 2 healthcare-08-00539-f002:**
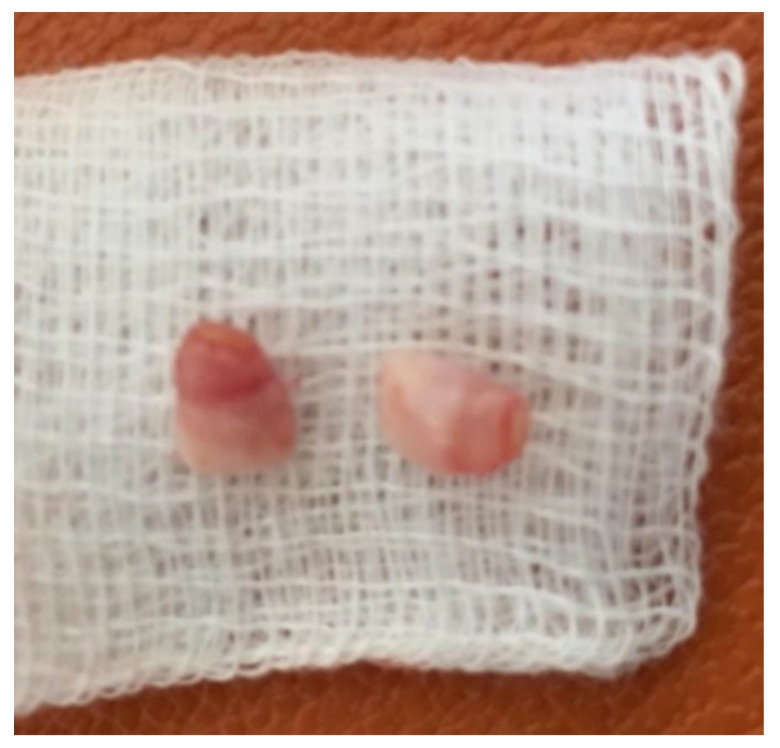
Extracted natal teeth.

**Figure 3 healthcare-08-00539-f003:**
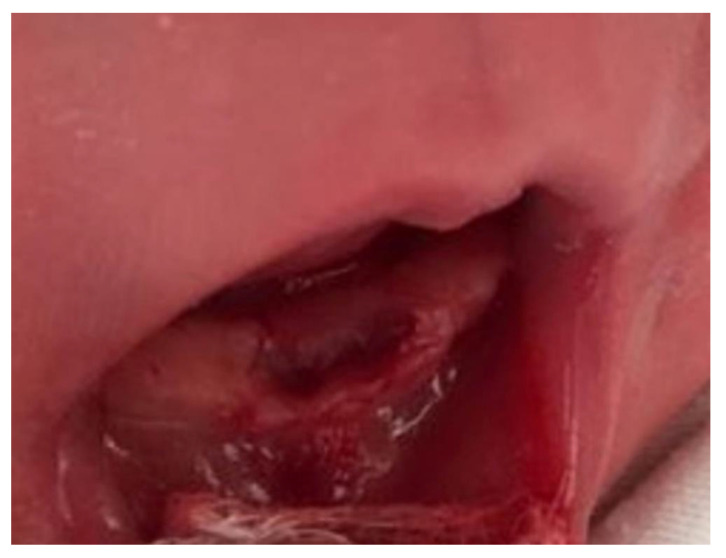
Postoperative site.

**Figure 4 healthcare-08-00539-f004:**
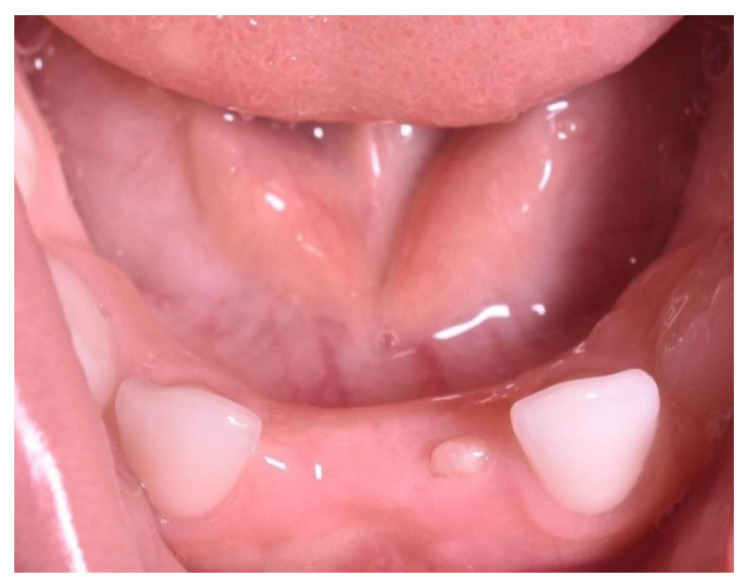
Clinical follow-up done after two years.

**Table 1 healthcare-08-00539-t001:**
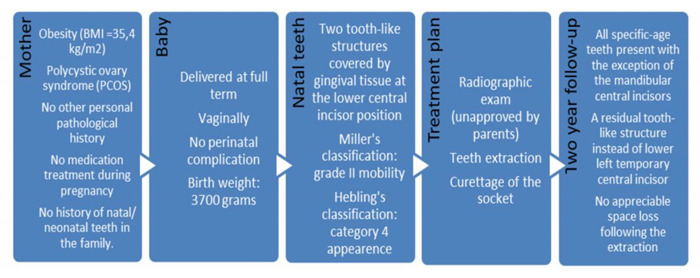
Summary of the case report.

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
