# Peer review of "Natal and Neonatal Teeth: A Case Report and Mecanistical Perspective"

_healthcare, 2020, doi:10.3390/healthcare8040539_

Round 1

Reviewer 1 Report

The authors describe a case of natal teeth that were treated by extraction without radiology investigation and later shown to be a part of temporary ones.

The case is interesting showing the importance of proper investigation before extraction.

The case data should be rewritten in more flow with addition of flow chart with data. The photographs are of pure quality. If radiology was performed later, it should be shown.

The review of the literature on the same topic with more interactive discussion should be added.

Author Response

Reviewer #1:

The authors describe a case of natal teeth that were treated by extraction without radiology investigation and later shown to be a part of temporary ones. The case is interesting showing the importance of proper investigation before extraction. The case data should be rewritten in more flow with addition of flow chart with data. The photographs are of pure quality. If radiology was performed later, it should be shown.

We summarized all the data from our case report in a new flowchart, as the reviewer kindly requested. We also improved the quality of the photos. As mentioned, written consent was obtained for the extraction and the participation in the study (attached for the Editorial usage at the beginning of the submission), but not for the radiographic exam. We added this aspect now as a limitation to our present study, at the end of the Discussion section. Thank you.

The review of the literature on the same topic with more interactive discussion should be added.

We improved our Discussion section by adding information after reviewing 6 new references regarding the gender differences, the differences in shape and size of natal/neonatal teeth and the presence/absence of hypoplastic enamel from natal/neonatal teeth, as you kindly asked.

Reviewer 2 Report

This manuscript presents a case of a new-born with two natal teeth. The authors emphasised the importance of proper diagnosis that includes a radiographic examination to manage these cases. the authors also provided an overview of the literature and theories regarding natal and neonatal teeth as part of the discussion. Overall, the manuscript reads well, the case is well presented and supported by references.

However, some points should be addressed to enhance the overall quality of the manuscript.

P2 L62-68: Mention that this classification is referred to as Helbing’s classification.

Given that there are many case reports and case series of natal and neonatal teeth in the literature. The authors should provide more information on the rationale of the study.

Was consent obtained from the parents for the case report?

Please provide higher quality intra-oral figures (especially Figure 3)

It would be useful to provide intra-oral radiographs.

P4 L108-112: To eliminate confusion for non-dental readers, replace 7.1 by (Lower lift temporary central incisor).

Author Response

Reviewer #2:

This manuscript presents a case of a new-born with two natal teeth. The authors emphasized the importance of proper diagnosis that includes a radiographic examination to manage these cases. The authors also provided an overview of the literature and theories regarding natal and neonatal teeth as part of the discussion. Overall, the manuscript reads well, the case is well presented and supported by references. However, some points should be addressed to enhance the overall quality of the manuscript. P2 L62-68: Mention that this classification is referred to as Helbing’s classification.

We mentioned that we categorized the natal/neonatal teeth using Helbing’s classification, as you kindly asked.

Given that there are many case reports and case series of natal and neonatal teeth in the literature. The authors should provide more information on the rationale of the study.

We added, at the end of the introduction section, the justification of our paper as the reviewer requested:

“Given the relative rareness of natal/neonatal teeth occurrence and the possible serious complications that may arise, the main purpose of the present paper was to present the case of a newborn baby with natal teeth in the mandibular anterior region. By presenting our case report and analyzing the available literature, we believe we emphasize the importance of a comprehensive and detailed oral cavity exam for each newborn baby. “

Was consent obtained from the parents for the case report?

Written consent was obtained for the extraction and the participation in the study (attached for the Editorial usage at the beginning of the submission), but not for the radiographic exam.

Please provide higher quality intra-oral figures (especially Figure 3)

We did improve the quality of our photos provided, focusing on figure 3. Thank you!

It would be useful to provide intra-oral radiographs.

As mentioned above and in the manuscript, written consent was obtained for the extraction and the participation in the study (attached for the Editorial usage at the beginning of the submission), but not for the radiographic exam. We added this aspect now as a limitation to our present study, at the end of the Discussion section. Thank you.

P4 L108-112: To eliminate confusion for non-dental readers, replace 7.1 by (Lower lift temporary central incisor).

We replaced 7.1 with lower left temporary central incisor, as the reviewer kindly suggested.

Round 2

Reviewer 1 Report

The manuscript was improved by the authors. The reviewer´s comments were well followed. The illustrations are improved. Flow chart helps readers to follow the case development. The pitfalls are nicely shown.

Reviewer 2 Report

The manuscript has improved. All my comments were addressed (apart from the quality of intra-oral figures) and I have no further comments.